# Extensive Variation in Thermal Responses and Toxin Content Among 40 Strains of the Cold-Water Diatom *Pseudo-nitzschia seriata*—In a Global Warming Context

**DOI:** 10.3390/toxins17050235

**Published:** 2025-05-09

**Authors:** Caroline Weber, Anna Junker Olesen, Robert G. Hatfield, Bernd Krock, Nina Lundholm

**Affiliations:** 1Natural History Museum of Denmark, University of Copenhagen, Øster Farimagsgade 5, 1353 Copenhagen K, Denmark; annakkolesen@gmail.com; 2Centre for Environment, Fisheries and Aquaculture Science, The Nothe, Barrack Road, Weymouth, Dorset DT4 8UB, UK; robert.hatfield@cefas.gov.uk; 3Chemische Ökologie, Alfred Wegener Institut-Helmholtz Zentrum für Polar-und Meeresforschung, Am Handelshafen 12, 27570 Bremerhaven, Germany; bernd.krock@awi.de

**Keywords:** arctic, climate change, domoic acid, intraspecific variation, temperature, trait variability

## Abstract

Phytoplankton are single-celled microorganisms with short generation times that may comprise high diversity in genetic and phenotypic traits, allowing them to acclimate to changes rapidly. High intraspecific genetic variation is well known in phytoplankton, but less is known about variation in physiological traits. To investigate variability and plasticity in genetic, morphological, and physiological traits of the toxigenic diatom genus *Pseudo-nitzschia* in a global warming scenario, we exposed 40 strains of the cold-water *P. seriata* to different temperatures (2 °C, 6 °C and 10 °C). The maximum growth rate and cellular toxin content showed extensive intraspecific variation, whereas morphological and genetic variation was minor. Thermal reaction norms showed a general increase in growth rate with increasing temperature; however, three distinct types of thermal responses were found among the 40 strains. All 40 strains contained toxins (domoic acid) in both exponential and stationary growth phase, and toxin content increased significantly with temperature. Most strains (>87%) contained measurable levels of domoic acid at all three temperatures. In conclusion, *P. seriata* shows extensive intraspecific variation in measured physiological traits like growth and toxin content, a variation exceeding the response of each strain to increases in temperature. Intraspecific variation in harmful species thus needs attention for the future understanding of food web dynamics, as well as the management and forecasting of harmful blooms.

## 1. Introduction

Phytoplankton account for approximately 50% of the global carbon fixation [1], and their production is directly coupled to the amount of energy transferred to higher trophic levels. Hence, what happens at the basis of the food web affects the rest of the food web. Polar as well as northern temperate waters are characterized by massive spring blooms. Diatoms often dominate these spring blooms, and, due to their heavy siliceous frustules, they are important components in the transport of carbon, nutrients, and silicon to ocean sediments [2,3].

As global air temperatures increase with climate change, a similar increase in oceanic temperatures is inevitable [4]. Temperature increases, however, vary across the globe, and the average temperature in the Arctic has risen almost twice the global average [5]. Warming of the ocean is expected to affect the ecosystem in several different ways, with the factors sometimes interacting with and reinforcing each other, sometimes working in opposite directions. The increase in temperature and light availability is predicted to result in an increase in primary production [6] and earlier spring blooms, thus possibly risking trophic mismatches [7,8]. Impacts on biodiversity and ecosystems are undoubtedly to be expected [9]. Furthermore, extreme events like marine heatwaves—long periods of abnormally warm seawater—are expected to become more frequent and intense [10] and expose organisms to temperatures above their tolerance, resulting in significant effects on marine ecosystems and biodiversity [11,12] and even causing blooms of toxic *Pseudo-nitzschia* [13,14]. Hence, understanding the response of phytoplankton, including diatoms, to increasing temperature is relevant.

Ecosystem functioning is closely linked to species diversity, and systems with a more diverse species composition are associated with higher productivity and stability [15,16]. Similarly, variation within species can lead to increased population growth as different phenotypic traits are able to complement each other, granting species a higher chance of surviving and adapting to environmental changes [17]. Phytoplankton have short generation times and oftentimes comprise high diversity in both genetic and phenotypic traits, allowing them to acclimate to changes rapidly [18,19]. Even though the ocean lacks apparent dispersal barriers, there is evidence for extensive genetic diversity between geographically different phytoplankton populations [20,21]. High genetic intraspecific variation is well known in phytoplankton (e.g., [22,23]), but fewer studies have explored intraspecific variation in physiological traits [24,25,26,27,28], and even fewer have investigated intraspecific variation in response to environmental changes [26,29,30,31]. The ability of a genotype to express different phenotypes across different environments is expressed as phenotypic plasticity and can be illustrated by reaction norm experiments, exposing genotypes to a range of a certain environmental factor.

*Pseudo-nitzschia* species are globally distributed diatoms known to form harmful algal blooms (HABs), as at least half of the species produce the neurotoxin domoic acid (DA) [32,33]. Domoic acid can be transferred to higher trophic levels, affecting endothermic animals [13,34,35,36,37]. Grazers such as copepods, krill, and filter-feeding bivalves function as vectors [38] but may also be affected by the toxin (e.g., [39,40]). This study focuses on the cold-water species *P. seriata* found in northern temperate and Arctic regions [41,42,43] where it occasionally forms HABs. Toxic strains of *P. seriata* have presently been found in Denmark, Iceland, Greenland, Scotland, and Canada [40,41,42,44], and incidents of accumulation of DA produced by *P. seriata* have been documented in sea scallops and shellfish from Canada [45,46] and blue mussels from Denmark [47]. In *Pseudo-nitzschia* species, as in many other phytoplankton species, genetic variation is relatively well known (e.g., [48,49,50]), whereas variation in physiological traits is scarcely investigated. To the best of our knowledge, studies include one to a maximum of five strains, and intraspecific trait variation in response to external factors like temperature has not been unequivocally explored so far [42,44,51,52,53,54]. The aim of the present study was to explore trait variation in coexisting strains of the cold-water diatom *Pseudo-nitzschia seriata*, exploring morphological, genetic, and physiological (growth and toxin content) traits in 40 strains as well as the trait plasticity in response to increasing temperatures (2 °C, 6 °C, and 10 °C). Intraspecific variation and trait plasticity are rarely studied in phytoplankton species and may be of particular relevance in harmful species for the future prediction and management of blooms and understanding of ecosystem dynamics.

## 2. Results

### 2.1. Growth Traits

Thermal reaction norms for all strains revealed extensive phenotypic variation among the 40 *P. seriata* strains, including large variation within the same temperature (Figure 1). Within each temperature, the variation in growth rates (day^−1^) varied between 8- and 12-fold; at 2 °C from 0.08 to 0.67 day^−1^, at 6 °C from 0.08 to 0.98 day^−1^, and at 10 °C from 0.10 to 0.91 day^−1^. The mean maximum growth rate increased with the temperature being 0.35 ± 0.12, 0.47 ± 0.19, and 0.50 ± 0.18 at 2 °C, 6 °C, and 10 °C, respectively. The mean maximum growth rates of the 40 strains at 2 °C were significantly lower than at both 6 °C and 10 °C (*p* < 0.005 and *p* < 0.0001, respectively), whereas the growth rates at 6 °C and 10 °C did not differ significantly from each other (*p* > 0.5) (Figure 1). Variation in growth rate of the strains in triplicate showed a standard deviation of 0.10 (day^−1^) at 2 °C, 0.07 (day^−1^) at 6 °C, and 0.13 (day^−1^) at 10 °C (Appendix A.

All strains were assigned a *type* of thermal reaction norm, describing the overall pattern of the individual strain growth rate in response to increasing temperature. Three main thermal reaction types (whether the growth rate increased or decreased when comparing 2 °C, 6 °C, and 10 °C) were detected (Figure 1 and Figure 2) in most of the strains (85%). The most common type, observed for 40% of the strains, showed a significantly increasing growth rate with increasing temperature (Type 1; Figure 2A), with the mean growth rate increasing 1.4-fold from 2 °C to 6 °C and 1.3-fold from 6 °C and 10 °C. The two other main types, ‘highest growth rate at 6 °C’ (Type 2; Figure 2B) and ‘growth rate highest at 10 °C, but similar at 2 °C and 6 °C’ (Type 3; Figure 2C), were also relatively common (30% and 15%, respectively). In Type 2, the mean growth rate at 6 °C was significantly higher than at 2 °C and 10 °C (1.7-fold and 1.5-fold, respectively). The remaining 15% of the strains exhibited thermal reaction norm types that were too rare to distinguish; thus, they were pooled as the ‘rest’ (Figure 2D). None of the types were significantly different from each other when comparing growth rates at 2 °C (*p* > 0.5). The growth rates at 6 °C were significantly higher in Type 2 than in Type 3 (*p* < 0.01), while the growth rates at 6 °C of Type 1 did not differ from Types 2 and 3 (*p* > 0.8). At 10 °C, the growth rate of Type 1 was significantly higher than in Type 2 (*p* < 0.01), whereas Types 2 and 3, and 1 and 3, were pairwise similar (*p* > 0.09).

The carrying capacity in stationary phase was statistically similar among all temperatures (~30.000 cells mL^−1^; *p* > 0.9) (Appendix A), and the mean lag phase did not differ significantly among temperatures (*p* > 0.1) (Appendix A), although both parameters comprised variation within each temperature.

### 2.2. Cellular Toxin Content

All 40 strains contained domoic acid. At the same temperature, cellular DA content varied considerably among strains, ~1100-fold, ~770-fold, and ~2670-fold at 2 °C, 6 °C, and 10 °C, respectively, in the exponential growth phase (Figure 3, Appendix A). The largest difference among strains was found at 2 °C (0.002–1.96 pg DA cell^−1^), while the highest DA content was measured at 10 °C (9.84 pg DA cell^−1^) (Figure 3, Appendix A). When pooling the results of all strains, a general trend of increasing toxin content with increasing temperature was seen in both growth phases (Figure 3) and most often with a significant correlation. Mean DA content (pg cell^−1^) at 10 °C was in both growth phases significantly higher than at 2 °C and 6 °C (*p* < 0.005). Of the strains, 35 showed the highest DA content at 10 °C. At 2 °C, the mean DA content in the exponential phase was significantly lower than at 6 °C (*p* < 0.01), but, in the stationary phase, there was no significant difference (*p* > 0.5). At each temperature, there was no significant difference in cellular toxin content between the exponential and stationary growth phase (*p* > 0.5). Cell volumes ranged from 789.7 µm^3^ to 2590.7 µm^3^. No correlation was found between maximum DA content (pg DA cell^−1^) and cell volume (µm^3^) (*p* > 0.3, Pearson’s r = 0.181).

### 2.3. Cellular Toxin Production Rates

In the exponential growth phase, the mean cellular DA production rates (pg DA cell^−1^ day^−1^) were significantly higher at 10 °C than at the two lower temperatures (*p* < 0.05; Figure 4), whereas, in stationary phase, mean DA production rates did not differ among temperatures (*p* > 0.1). Noteworthily, DA production rates varied considerably in both growth phases, especially at 10 °C (Figure 4). Mean cellular DA production rates of the two growth phases were not different from each other at each temperature (*p* > 0.1). Correlation tests in the exponential phase between the maximum GR and either the maximum DA content or the toxin production rate revealed no significant correlations (Appendix A).

### 2.4. Morphological Diversity

Prior to the experiment, intraspecific variation in the four morphometric characters was compared with previous descriptions of *P. seriata*, *P. australis* and *P. obtusa* in Hasle and Lundholm [55] (Appendix A). The range (min./max.) of the characters’ cell width, density of striae, and poroid density agreed with the accepted description of *P. seriata*, whereas the density of fibulae for most strains agreed with the description, but some strains had a slightly lower fibula density (Appendix A). When comparing especially *P. seriata* and *P. australis* data, it was evident that the ranges of most of the characters (cell width, fibulae density, and striae density) were overlapping, confirming their close evolutionary relationship, whereas poroid density was consistently lower in *P. australis*.

In the strains of *P. seriata* examined, the number of rows of poroids was usually arranged in two outer and two inner rows of poroids (expressed as 2 + 2 poroid rows). However, in some strains, some of the cells contained only two outer rows and one inner row of poroids (expressed as 2 + 1 poroid rows). In total, 52% of the strains had 2 + 2 poroid rows (morphotype 1; Figure 5), whereas the rest had a combination of 2 + 1 and 2 + 2 (morphotype 2; Figure 5).

### 2.5. Genetic Diversity

The most notable variation in the sequences generated (18S, ITS, and D1-D2 of 28S of the ribosomal DNA operon) was a microsatellite insertion of four bases (GCCC repeat) in the ITS1 region in only strain A−10 (Figure 6). The integrative genomic viewer (IGV) highlighted that 74% of sequences generated for the strain had this insertion. Analysis by RNAfold 2.5.1 indicated that this would affect the secondary structure of ITS1. The nanopipe highlighted three single-nucleotide polymorphisms (SNPs), which were confirmed by KMA and IGV (Figure 6). The first was at position 1422 of the aligned sequences, as a transversion from cytosine to arginine in 18S in strain F1-N only (Appendix A), with almost all sequences (90%) generated for this strain having this mutation present but was not seen in any other strains. The second was an SNP at position 2239, in the ITS2 region, with a transition from thiamine to cytosine. This substitution was common, with a third of strains found to have both variants present as a degeneracy when interrogated by IGV and KMA. However, only three strains (A6-N, A8-N, and C7-N) had the cytosine in higher abundance than thiamine. It is noteworthy that this degeneracy was observed in the reference sequence Nissum3 from Denmark AY257841 (Appendix A). The third SNP was a transition from guanine to adenine observed in the D2 region of the 28S gene only in strain B4-N at position 2946. Analysis by RNAfold 2.5.1 highlighted that none of these three SNPs would result in changes to the amino acid sequence secondary structures. In addition, deletions were observed at 11 sites of the aligned sequences. KMA analysis highlighted these as unreliable, and locations of these deletions were noted as either homopolymer or tandem repeat regions, known to be problematic for nanopore sequencing [56]. It is therefore suspected that these are a result of sequencing errors in the raw reads manifesting in the consensus sequences of samples and were not investigated further.

## 3. Discussion

Extensive intraspecific variation in *P. seriata* was found in all the measured physiological traits (growth, DA content, and DA cellular production rate) (Figure 1, Figure 4 and Figure 5). This pronounced variation emphasises the need to carefully consider general conclusions based only on single- or few-strain studies, as also noted by Godhe and Rynearson [19]. Intraspecific variation is important to consider in several contexts, as it may be even larger than interspecific variation [26,28]. Like in the current study, a study of 24 isolates of the cyanobacterium *Cylindrospermopsis raciborskii* from the same water sample also revealed substantial variation among isolates [25]. None of the 24 strains of *C. raciborskii* had identical growth rates, toxin quotas, or morphological traits. Similarly, high intraspecific variation in physiological traits was found among five strains of *Skeletonema marinoi* [57] and among other taxa from a single location [24,29,58,59].

The strains used in this study all originated from the same water sample, thereby representing a minimum of the ‘true’ intraspecific variation. Ryderheim and Kiørboe [60] found that studies examining the intraspecific variation in coexisting strains are few; hence, more studies including multiple strains originating from the same population are needed. Among the studies, they found that most studies included five strains or fewer and that an average of 2.68 traits were quantified in the studies [60]. The current study included a large number of strains and traits, but with compromises to the number of technical and biological replicates. However, a sort of replication was conducted as measurements were performed by measuring the same strain over several growth cycles.

If considering different spatial and temporal scales and different environmental conditions, the variation in a species is probably even larger. However, a study reviving 18 strains of the dinoflagellate *Pentapharsodinium dalei* from three sediment core layers (representing a time span of 90 years) points in a different direction: while growth rates differed significantly among strains (ranging 0.02–0.65 day^−1^), they did not differ significantly with time [61]. In contrast, a study exploring population genetic markers of the same organisms revealed changes in diversity over a time span of almost a century, with shifts in subpopulations coinciding with changes in hydrographic conditions [62]; hence, adaptation of the strains would most likely be reflected when exploring responses to certain environmental parameters.

Several phytoplankton species have resting stages that may increase the variation in species even more [63]. Phenotypes that are not optimal in time periods with certain environmental conditions may survive as resting stages and provide an inoculum when the environment is suitable for that particular phenotype. This may help to explain the long-term survival of aquatic protist species [19,62,64]. Resting stages have not yet been confirmed in *Pseudo-nitzschia* species; however, sediment cores indicate the possibility [65].

Geographical origin has also been found to be important for the intraspecific diversity. A total of 32 strains of *Microcystis aeruginosa* isolated from 12 lakes exhibited a three-fold variation in growth rates (0.17 to 0.46 day^−1^) [66], and 28 strains of the diatom *Leptocylindrus danicus* from eight locations along the east coast of Australia showed growth rates ranging from 1.25 to 1.65 (day^−1^) [28]. Finally, 15 strains of the dinoflagellate *Alexandrium ostenfeldii* from the same creek but in two distinct years (2015 and 2016) displayed growth rates varying three-fold, from 0.12 to 0.39 (day^−1^) [27].

Ultimately, this all shows that phytoplankton exhibit high intraspecific variation in growth traits even among strains sampled at the same time and at the same place, but that diversity may be even larger when including variation on spatial and temporal scales.

The increase in growth rate in *P. seriata* with increasing temperature is in accordance with a well-documented principle that a moderate temperature increase results in a higher growth rate in phytoplankton species [67], as well as in diatoms [68,69,70], with exceptions in the cold-water diatom *Porosira glacialis* [71]. The fact that the growth of *P. seriata* was positively affected by increasing temperature but that the mean growth rate did not differ between 6 °C and 10 °C (Figure 1) agrees very well with a previous study on a single Canadian arctic strain of *P. seriata* [72]. The mean growth rate at 2 °C (0.37 day^−1^), 6 °C (0.49 day^−1^), and 10 °C (0.46 day^−1^) found by Stapleford and Smith [72] agrees almost perfectly with the mean growth rate of the present experiment at the same temperatures (0.35 day^−1^ at 2 °C, 0.47 day^−1^ at 6 °C, and 0.50 day^−1^ at 10 °C). In another study, a *P. seriata* strain sampled from sea ice was exposed to eight different temperatures ranging from −1.6 °C to 15 °C. The results found growth between −1.6 °C and 12 °C [73]. Likewise, Hansen et al. [42] found that neither of the two Greenlandic strains of *P. seriata* grew well at 15 °C. However, two Scottish strains of *P. seriata* successfully grew at 15 °C [41].

In conclusion, *P. seriata* comprises a considerable intraspecific variation in physiological response to temperature changes and has, as a species, a broad temperature range (of at least −1.6 °C to 15 °C), an optimum temperature of around 6–10 °C, and an inherent huge intraspecific variation even within a single location.

Temperature is a fundamental driver of physiological processes and important for growth and production. Phytoplankton species exist at specific temperature ranges restricted by upper and lower lethal temperatures [74], and thermal tolerance curves (reaction norms) follow a skewed bell shape with an optimum temperature and reduced growth rate at higher and lower temperatures [75,76]. Often, species are found at lower temperatures than their optimum; hence, increasing temperatures may result in increased growth rates [67,68]. Altogether, the *P. seriata* strains showed significantly increasing growth rates with increasing temperature, although with no significant difference between 6 °C and 10 °C (Figure 1). The thermal response types (Figure 2) reveal phenotypic plasticity and diversity in the growth response, demonstrating that *P. seriata*, as a species, seems to be resilient to temperature increases. The overall thermal norm of *P. seriata* is the sum of all types of thermal responses and does not represent the ‘true’ response of (most) single strains. The species, as such, will hence survive temperature increases, but the composition of the population will change when temperature changes.

More than 100 single cells or chains resembling *P. seriata* were isolated in the present study, of which ~70 cultures were established, hence an isolation success of ~70%. This implies additional intraspecific variation due to the isolates, which were not successfully cultured, e.g., because they did not grow well under the offered laboratory conditions [77]. Similarly, Tesson et al. [78] had an isolation success of *P. multistriata* of 66.8%, while isolation success for *Ditylum brightwellii* ranged from 38% to 96% [24], emphasizing this as a general challenge. Methods used for isolating and culturing microalgae include both deliberate and unintentional choices, all potentially affecting the results. The isolates not surviving in culture hence represent an inherited bias favouring cultivable strains [79]. This hurdle is not easy to overcome, but it is an important factor to address when interpreting data.

The observed responses reflect acute thermal stress rather than longer-term acclimated performance, and they are relevant, e.g., for responses to marine heatwaves, which are expected to become more frequent and have been causing severe toxic blooms of *Pseudo-nitzschia* [17,18]. Furthermore, the approach of exploring acute thermal stress allows us to assess the baseline physiological variation present in the population, as it highlights how different strains respond to abrupt, but environmentally realistic, changes in temperature. A set-up in a long-term evolutionary context exploring long-term thermal adaptation is not the scope of the present study.

All 40 strains of *P. seriata* contained DA, but with a significant variation in cellular amount. Similarly, huge variations in toxin content have been reported among 15 strains of *A. ostenfeldii* (~74-fold) [27]. Even though all 40 strains of *P. seriata* contained DA, many toxigenic species are known to comprise both toxic and non-toxic strains, e.g., only one of five strains of *P. simulans* produced DA [52], and 19 of 21 strains of *P. multiseries* produced DA, with DA content varying considerably (4.4 to 2220 ng DA mL^−1^) [80]. The strains used in the current study were relatively young when DA was measured (<1 year since isolation). For the 21 *P. multiseries* strains, 11 were <1 year old, and the rest were 1–6 years old [80]. It has been shown that *Pseudo-nitzschia* cultures tend to decrease in cellular DA content over time in culture [81,82]. The reasons could be decreasing cell length through asexual reproduction [82,83] and/or a lack of exposure to toxin-inducing factors like grazing copepods [40,84]. However, Evans et al. [80] found older strains to be toxic and toxin levels relatively high. A regain of toxicity has also been seen in cells after sexual reproduction [83].

Increasing DA content in *P. seriata* with increasing temperature was similar to that of a single strain of *P. australis,* which showed an increasing cellular DA production rate (pg DA cell^−1^ day^−1^) with increasing temperature [85]. However, whereas a clear positive correlation between DA production rate and temperature was detected in *P. australis*, no clear correlation between cellular DA production rate and temperature was detected in the current study, even though a trend of increasing cellular DA production rate with increasing temperature was found in the exponential phase (Figure 4). In both growth phases, strains with negative cellular production rates were observed (Appendix A). A negative cellular production rate may reflect cellular DA content being ”diluted” when the strains grow rapidly, or at least that the growth rate overrides the cellular DA production rate. Another explanation could be the DA leaking from the cells and ending up in the water as dissolved DA. Unfortunately, dissolved DA was not measured in the current study, but dissolved DA has previously been found to account for only a minor fraction of total DA in *P. seriata* [40].

Similar to our results on the growth rate versus toxin content, Wilson et al. [66] found no correlation between the growth rate and toxin content in 29 toxic strains of the cyanobacterium *Microcystis aeruginosa*. This would suggest no obvious trade-off between the growth rate and toxicity in both *P. seriata* and *M. aeruginosa*. In contrast to this, the negative relationship between the growth rate and toxin content in 15 strains of the dinoflagellate *A. ostenfeldii*, suggested a trade-off between growth and toxin production [27]. Even though the present study did not show any trade-off between growth and toxin content or production, the production of toxins most likely does come at a cost, as one would otherwise expect all algal species to produce toxins. The costs of DA production in *P. seriata* have recently been demonstrated, where the cells of *P. seriata* with high DA cell quota correlated with a decrease in growth rate—a clear trade-off—with similar costs for all strains [40]. It has been suggested that the difficulty in measuring the cost of toxin production in *Pseudo-nitzschia* could be caused by DA levels being too low and growth being unlimited by external factors in laboratory conditions to sufficiently imply measurable differences in growth rates [84], e.g., under non-limiting conditions of energy (light), which might explain why we did not find a trade-off between the growth rate and toxin content. We found no difference between higher DA production rates in the exponential and stationary growth phases at any temperature. These results are in contrast with Cochlan et al. [86], who found significantly higher DA production rates in the exponential than in the stationary growth phase in *P. australis*. This may represent a difference between species, as some *Pseudo-nitzschia* species only produce DA during the stationary phase, while others like *P. australis* and *P. seriata* also produce during the exponential phase [32,87].

The morphological characters assessed (cell width and densities of fibulae, striae, and poroids) all seem to be relatively stable intraspecific characters, except for a slightly broader range for fibula density than proposed by Hasle and Lundholm [55] (Appendix A). This supports previous studies finding these characters stable among strains of *Pseudo-nitzschia* species [42,88,89]. Regarding the number of rows of poroids, we found two morphotypes among the strains; one group always had 2 + 2 rows of poroids (two outer rows of larger poroids and two inner rows of smaller poroids), while the other group included cells with 2 + 2 rows of poroids and others with 2 + 1 (Figure 5). This agrees with observations of *P. seriata* from Scottish waters [41]. Apart from intraspecific variation in the trait, it also varies with temperature, as a tendency of reduction in rows of poroids (from four to two) with higher temperature [42]. The three species *P. seriata*, *P. australis,* and *P. obtusa* generally overlap in their characters, including the number of poroid rows, making correct identification challenging (Appendix A) [55].

The genetic diversity among the strains of *P. seriata* highlighted three SNPs (two transitions and a transversion compared with reference sequences) (Figure 6). An analysis of secondary structures associated with these variants using RNAfold 2.5.1 found no change in the amino acid secondary structure once transcribed. It is therefore unlikely that they would have caused any disruption in ribosomal performance. The transition of thymine to cytosine in position 2239, creating the Y degeneracy, was previously characterized for Nissum3, accession: AY25784 [90] (Appendix A). The variable level manifestation of this degeneracy between strains potentially indicates the interbreeding of organisms with and without this mutation, as has been observed in other such anomalies in studies of *P. pungens* and *P. multiseries* [91,92].

The microsatellite observed in the ITS1 region of strain A10-N was a significant variation and would have resulted in a change in amino acid secondary structure. As this was not observed in any other strains and no examples of this could be identified on the NCBI database, it is thought to have been an isolated event, possibly due to replication slippage. Previous studies have highlighted the abundance of microsatellite insertions and their suitability for genetic population applications for *Pseudo-nitzschia*, albeit with a different species [93]. However, it is possible that this could be caused by the replication of a PCR error that cannot be proven without further investigation.

Phenotypic plasticity coupled with intraspecific variation among phenotypes in a species acts to buffer the response to environmental fluctuations [19]. The higher the intraspecific variation, the higher the likelihood of a species’ survival chances. For the diatom *Skeletonema marinoi*, the highest primary production under salinity stress was found in a mixed culture of 20 strains compared to monocultures or a mixture of five strains, emphasizing the importance of genetic diversity for ecological performance [57]. Hence, as a species, *P. seriata* seems to be resilient to future temperature increases due to the inherent diversity. A previous study showed something similar, as revived *Pentapharsodinium dalei* from sediment cores revealed two genetic subpopulations that shifted in dominance through time according to environmental changes [62].

Arctic diatom species have, in general, been shown to have higher growth rates at temperatures higher than where they are naturally found [69,70], suggesting that other factors (e.g., predators or competition) are co-defining the range of the species. Ultimately, Arctic species might be capable of surviving a temperature increase, but warming may lead to new species arriving, adding to the competition among species. In addition to biological factors (e.g., competition, grazing) affecting species ranges, climate variables often interact. Even though the current study indicates temperature as having a positive effect on the growth rate and toxicity of *P. seriata*, interactions of, e.g., temperature, lower pH, and less nutrients may lead to a different outcome. In an experiment testing the interactive effect of temperature, CO_2_, and salinity on phytoplankton communities at two sampling locations in the Arctic Ocean, they found that the three climate variables affected community structure differently and that the variables even had different effects on composition at the two locations [94]. Similarly, in an experiment on *P. multiseries*, ocean warming, and acidification, the effect of pCO_2_ levels depended on temperature levels [95].

Responses to future climate conditions may encompass responses to several of both abiotic and biotic factors and can be difficult to predict. An elegant study on *Skeletonema marinoi* from the Baltic Sea sediment, where strains were resurrected 60 years back in time showed; however, that present-day *S. marinoi* strains altogether had adapted to higher sea temperatures by increasing their temperature optima by 1 °C when compared to 60-year-old strains [96]. This demonstrates that, on top of intraspecific diversity and plasticity, the survival of species to future climate changes will depend on the longer-term adaptation of naturally occurring strains to increasing temperatures. The results of the present study underline the importance of considering intraspecific variation as a basis for future studies on the survival and adaptation of a species. Still, experiments including a combination of environmental factors, as well as the intraspecific variation, are needed to determine how *P. seriata*, other *Pseudo-nitzschia* species, and other harmful species will react to future changes in the ocean. In the last 30 years, there has been a 7-fold increase in global observations of *Pseudo-nitzschia* coupled with a similar increase in toxic events involving domoic acid [97]. Altogether, and despite all qualifications, our results may suggest that global warming could result in a higher risk for toxic *Pseudo-nitzschia* blooms.

An increased occurrence of HABs of *Pseudo-nitzschia* as well as of other species may pose a risk to public health and lead to important economic losses [98,99]. If higher toxicity with increasing temperature is a general trend among toxigenic species, current action thresholds might underestimate the potency of blooms, thus posing a greater risk to shellfish farms and fisheries [38,100,101]. Moreover, the increasing toxicity of blooms could result in higher accumulations of toxins in the food web, potentially having fatal consequences for higher tropic levels [102].

## 4. Conclusions

Testing the physiological response of 40 strains of *P. seriata* to three temperatures (2 °C, 6 °C, and 10 °C) revealed substantial variation in growth rate and DA content, which emphasizes the importance of considering intraspecific variation when interpreting physiological responses to environmental factors. Results indicate higher growth potential and higher toxin content at 10 °C, potentially posing a risk of future HABs occurring in warming waters. On the contrary, morphological characters and genetic variation were rather stable parameters. This is interesting, as the variation in physiological characters is likely affecting ecosystem functioning directly compared to variation in genetics and morphology. Currently, little is known about intraspecific variation in phytoplankton on a broader scale, especially on physiological variation, which emphasizes the problems in the conclusions on the physiology of species depending on one or a few strains. Based on our results, it is likely that variation within other species can also be significant. This calls for further investigation into possible variations in the physiological response of other phytoplankton species. On top of that, the interaction of multiple environmental factors (e.g., temperature, salinity, pH, and nutrient availability) may complicate predictions about a species’ response to future change. Hence, future studies investigating intraspecific variation under multiple stressors could help better predict the ecological impact of a changing climate.

Intraspecific phenotypic variation might further complicate, e.g., the modelling of bloom events and thus over- or underestimate the risk, making the forecasting of HAB events tricky. The monitoring and risk management of harmful species like *Pseudo-nitzschia* become less straightforward when traditional methods, such as cell counts, likewise over- or underestimating the potency of a bloom event, result in ineffective resource allocation. Furthermore, our understanding of food web interactions and thus ecosystem dynamics may be hampered if we do not consider the extensive intraspecific variation and plasticity that species harbour [103]. Hence, it is of importance to consider intraspecific variation in several aspects of aquatic research involving phytoplankton traits.

## 5. Materials and Methods

### 5.1. The Strains: Isolation and Cultivation

In total, 100 cells or chains of *P. seriata* were isolated from a water sample from Skovshoved Harbour, Denmark (55°45′40″ N 12°36′0″ E), April 2020. Using a light microscope (Olympus CKX53), single cells/chains resembling *P. seriata* were isolated using a micropipette and placed in 96-well plates containing L1-medium based on filtered seawater [104]. Of the 100 isolates, 70 strains were successfully cultured and maintained in a 4 °C temperature-controlled room with a light/dark cycle of 16:8 at a light intensity of ~110 µmol photons m^−2^ s^−1^. Forty randomly chosen strains were used for the experiments within a year after isolation to reduce effects of culturing (see, e.g., the study of [77]). The experiment ran from December 2020 till July 2021; hence, strains were in culture for 7–14 months before the experiment.

### 5.2. Morphological Assessment and Species Identification

Before the experiments, approximately 15 mL samples of all strains were fixed using Lugol’s solution (final concentration 3%) and rinsed following [90]. A drop was applied on carbon-coated copper grids, dried, and assessed using a transmission electron microscope (TEM; JEOL 1010, Tokyo, Japan). Using TEM micrographs, cell width and length, densities of fibulae, striae, and poroids and number of poroid rows in a stria were assessed and compared to Hasle and Lundholm [55] for species confirmation (Appendix A). In total, 28 strains were assessed in detail with all morphometric characters from a minimum of three valves per strain. In total, 39 strains were identified as *P. seriata* genetically; the one strain not genetically identified was identified morphologically. Cell volumes (µm^3^) were calculated using the following equation, as proposed by Ayache et al. [105]:(1)$$\mathrm{Cell}\;\mathrm{volume}=\frac{1}{2}a\times b^{2}$$

With a representing cell length and b representing cell width.

### 5.3. Experimental Set-Up

For each of the 40 strains, inocula of experimentally growing batch cultures with an initial concentration of ~1300 cells mL^−1^ were transferred to 220 mL flasks and grown at three different temperatures under the conditions mentioned below. Maximum growth rate, cellular toxin content and cell density were followed until stationary phase. The experiment was conducted in temperature-controlled chambers set to 2.0 °C ± 0.6, 5.2 °C ± 0.4, and 9.3 °C ± 1.2 (referred to as 2, 6, and 10 °C) (Appendix A). The temperature was monitored with temperature loggers (HOBO TidbiT v2 Temp Logger, Onset, Mass, USA) submerged in water volumes equivalent to the volume in the experimental flasks. Temperatures were chosen based on (1) water temperature when sampled (~5 °C), (2) projections that sea surface temperatures will increase with up to 4.4 °C based on IPCCs models [106], and (3) the fact that *P. seriata* is found in Arctic and northern temperature waters and has been isolated from sea ice [72].

The flasks were placed at a light intensity of ~110 µmol photons m^−1^ s^−1^ using cool white LED light (Perel, Gavere, Belgium). After each subsampling, flasks randomly shifted position to counteract potential advantages or disadvantages of certain locations. The strains were not acclimatised to the temperature prior to the experiment, because the intention was to explore the inherent physiological plasticity in response to fluctuating conditions like those in coastal habitats and during, e.g., marine heatwaves. To maximise the number of strains examined, all were run in single replicates, but, for exploring potential in-strain variations, three strains (A9-C, C10-N, and F12-N) were run in triplicate. All strains were quantified using both fluorescence and cell counts. The cells were grown in L1-medium [104] but using 2 mL of silicate solution (Na_2_SiO_3_) to ensure silicate repletion and based on filtered seawater with a salinity of 30 and a pH of 7.9.

### 5.4. Sampling for Cell Densities

Relative fluorescence unit (RFU) of all flasks was measured approximately every second day using 1.5 mL of well-mixed sample and a fluorometer (Triology, Turner Designs, San Jose, CA, USA). Afterwards, the sample was counted using Sedgewick rafters, counting either a minimum of 400 cells or 200 squares using a light microscope (Olympus CKX53, Olympus Corp., Tokyo, Japan).

### 5.5. Toxin Harvesting and Analyses

Subsamples (45 mL) for cellular toxin content were sampled four times during the experiment of each strain: twice during exponential growth phase (referred to as ex1 and ex2, 24–48 h between sampling) and twice during stationary phase (referred to as sta1 and sta2, 24–48 h between sampling). The subsamples were centrifuged for 15 min, at 4 °C at a speed of 1811× *g* (centrifuge model 5810 R, Eppendorf). The supernatant was discarded, and the pellets were stored at −22 °C until further analysis. A 300 µL 1:1 mixture of methanol and 0.3 M acetic acid was added to the pellets. The samples were vortexed for 2 s and transferred to a 2 mL cryotube (Sarstedt micro-tube, Nümbrecht, Germany). To disrupt the cells, Lysing Matrix D beads were added till they covered the bottom of the tube. Afterwards, the tube was put in a FastPrep (Thermo BIO 101, FastPrep FP120, Illkirch, France) at a speed of 6.5 m s^−1^ for 45 s and centrifuged for 5 min at 16,100× *g* (Centrifuge 5415 R, Eppendorf). The supernatant was transferred to a filter tube (Ultrafree MC HV, Durapore PVDF 0.45 µm, Merck Millipore, Eschborn, Germany) and centrifuged. The filtered samples were stored in a sealed glass vial. DA was determined using liquid chromatography (LC 1100 Chromatograph, Agilent, Waldbronn, Germany) coupled with tandem mass spectrometry (LC-MS/MS) (API 4000 QTrap, Sciex, Darmstadt, Germany) as described in Krock et al. [107].

### 5.6. Molecular Data

The strains were analyzed using the nanopore sequencing technology. A minimum of 15 mL of each strain was sampled and centrifuged at 4 °C at 1811× *g* for 20 min (centrifuge 5810R, Eppendorf). The supernatant was discarded, and two drops of 2% acidic Lugol’s solution were added to the pellet. The samples were stored at 4 °C until further analysis. DNA extraction, amplification of rDNA by PCR and nanopore sequencing were performed as described in Hatfield et al. [108]. To avoid reference bias, a de novo approach was adopted for the generation of consensus sequences. Specifically, NGSpeciesID (v0.1.1.1) was used for clustering [109], and spoa (v4.07) generated consensus sequences, which were then polished using Medaka (v1.2.4) [101,110]. Variant calling was performed using Nanopipe, KMA, and integrative genomic viewer (IGV), with the latter used to visually interrogate the consensus sequences [111,112,113]. Observed anomalies were submitted for assessment of secondary structure sequences using RNAfold 2.5.1. via Vienna RNA Websuite [114]. In total, 52 strains were successfully analyzed, and 39 of the 40 strains used in the experiments were successfully characterized.

### 5.7. Data Analyses and Statistical Analyses

Maximum growth rates (day^−1^) were calculated for each strain at each temperature using the following formula:(2)$$\mathrm{GR}=\ln\left(\frac{N_{t2}/N_{t1}}{t_{2}/t_{1}}\right)$$
where N_t1_ and N_t2_ corresponds to cell numbers at times t_1_ and t_2_.

A type of thermal reaction norm for growth rate was determined for each strain by comparing growth rates at the three temperatures (e.g., growth rate increased with temperature or growth rate was unaffected by temperature). Carrying capacity was determined as the highest cell density a strain reached in stationary phase.

Cellular DA content (pg DA cell^−1^) was calculated by dividing the total toxin content of a sample with the respective cell concentration. The cellular DA production rate (pg DA cell^−1^ day^−1^) was determined in both exponential- and stationary growth phase according to Tong et al. [115] using the following equation:(3)$$\mathrm{Cellular}\;\mathrm{toxin}\;\mathrm{production}\;\mathrm{rate}=\frac{{C_{2}T}_{2}-{C_{1}T}_{1}}{\left(\frac{C_{2}-C_{1}}{\ln(C_{2}-C_{1})}\right)\times (t_{2}-t_{1})}$$

With C as cell abundance (cells mL^−1^) at the first (C_1_) and second (C_2_) sampling and T as cellular toxin content (pg DA cell^−1^) at the first (t_1_) and second (t_2_) sampling. The calculations were based on the cellular DA content only to reduce amount of toxin analyses, as previous studies have shown very minor amounts of dissolved DA in studies of *P. seriata* [40,42].

All statistical analyses were conducted using the computer programme GraphPad Prism 8 (version 8.4.2) using a significance level of 0.05. Growth and toxin data were tested for statistical significance using one-way ANOVA followed by Tukey’s multiple comparisons test. It was tested whether GR and toxin production and GR and toxin content correlated by computing the *P* value and Spearman correlation coefficient (Spearman’s r) for each comparison.

## Figures and Tables

**Figure 1 toxins-17-00235-f001:**
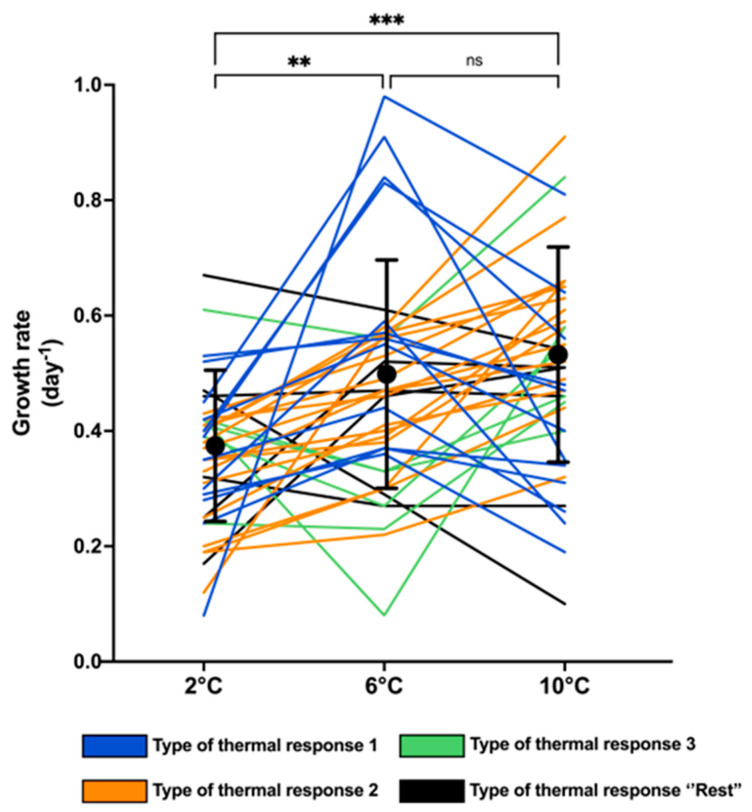
Overall phenotypic variation in growth rates (day^−1^) including mean and standard deviation (shown as black dots with error bars) of 40 strains of *Pseudo-nitzschia seriata* at temperatures 2 °C, 6 °C, and 10 °C. Each line represents a strain, and the colour of the line indicates the type of thermal response of the strain. Significant differences in mean values between temperatures are shown as ns = no significance, ** <0.01 and *** <0.0005.

**Figure 2 toxins-17-00235-f002:**
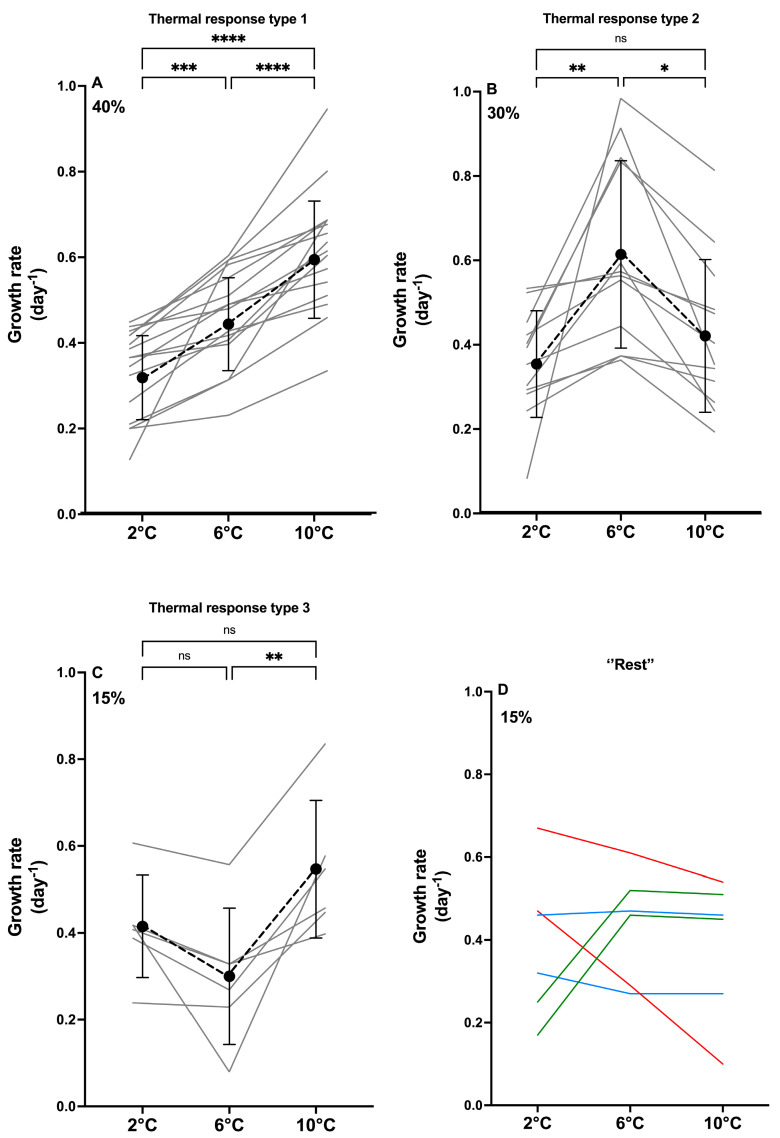
Phenotypic variation in growth rate based on data from 40 strains split into types of thermal reaction norms, revealing three different types (**A**–**C**) and the remaining strains (**D**). Percentage of strains exhibiting each type is shown upper left corner. (**A**–**C**) show thermal reaction norms for the strains following the assigned type as grey lines, and mean GR ± SD indicated by a black dot with error bars. (**D**) Strains, for which too few exhibit the same type to test statistically. Colors in graph (**D**) indicate strains with similar responses to temperature. Significant differences between temperatures are shown as ns = no significance, * = 0.05, ** < 0.005, *** < 0.001 and **** < 0.0001.

**Figure 3 toxins-17-00235-f003:**
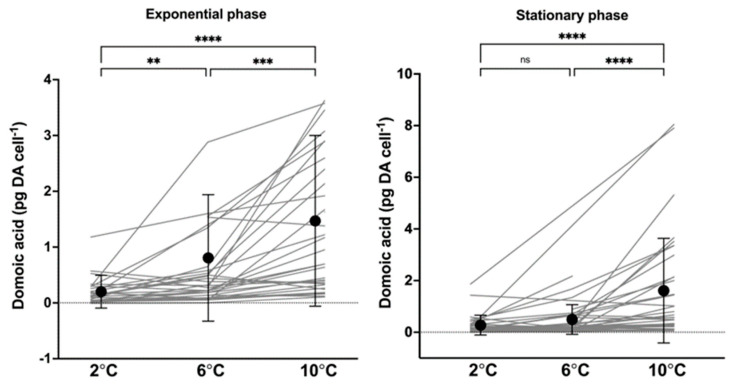
Domoic acid content (pg cell^−1^) in the exponential and stationary growth phase at temperatures 2 °C, 6 °C, and 10 °C. Dots with error bars represent the mean (±SD) at each temperature. The grey lines each represent a strain and indicate the progression of domoic acid content with temperature. Significant differences between temperatures are shown as ns = no significance, ** < 0.01, *** < 0.001 and **** < 0.0001.

**Figure 4 toxins-17-00235-f004:**
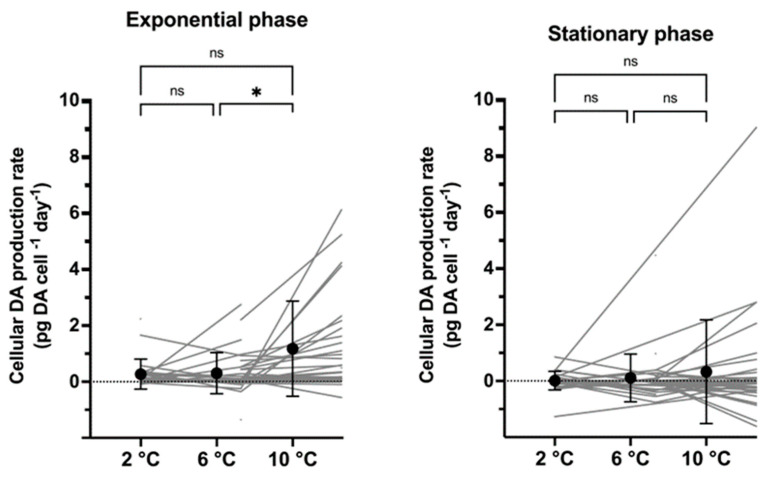
Cellular domoic acid production rate (pg DA cell^−1^ day^−1^) in exponential (**left**) and stationary growth phase (**right**) at three different temperatures. Significant differences between temperatures are shown as * < 0.05, ns = no significance.

**Figure 5 toxins-17-00235-f005:**
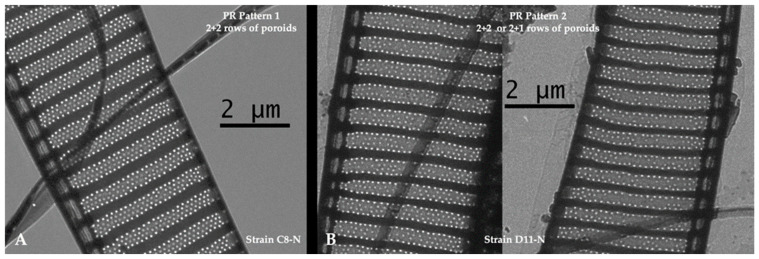
Micrographs of *Pseudo-nitzschia seriata* showing the two morphotypes of poroid row pattern in 28 strains: (**A**) morphotype 1 with a 2 + 2 poroid row pattern; (**B**) morphotype 2 with 2 + 1 or 2 + 2 poroid row pattern. Scale bars are 2 µm.

**Figure 6 toxins-17-00235-f006:**
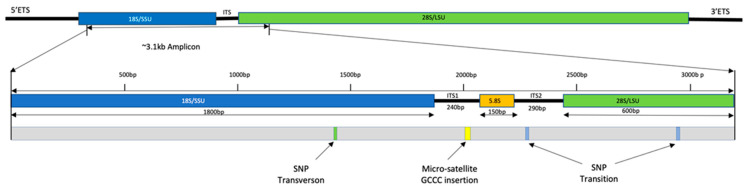
Intraspecific genetic variations observed in ribosomal DNA across 40 strains of *Pseudo-nitzschia seriata*. The small subunit (SSU/18S) is colored blue; the internal transcribed spacers one and two (ITS1&2) are colored black; the RDN58 gene (5.8S) is colored yellow; and the large subunit (28S/LSU) is colored green.

## Data Availability

The original contributions presented in this study are included in this article and the Appendix A. Further inquiries can be directed to the corresponding author.

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
