# Peer review of "Extensive Variation in Thermal Responses and Toxin Content Among 40 Strains of the Cold-Water Diatom Pseudo-nitzschia seriata—In a Global Warming Context"

_toxins, 2025, doi:10.3390/toxins17050235_

Round 1
Reviewer 1 Report
Comments and Suggestions for Authors
This manuscript studied the variation and toxin content of the cold-water diatoms in response to temperature. The results indicate that extensive intraspecific variation of P. seriata occurred in measured physiological traits, a variation exceeding the response of each strain to increases in temperature. Interestingly, the relevant results support that global warming may increase the risk of toxic diatom blooms. This manuscript is fully discussed. I recommend it to be published after a revision.
Line 16 and Fig. 1. What are the differences for each type of thermal response?
Figure 5. The scale bar in Figure 5B is not clear.
The discussion section. The authors need to merge identical viewpoints, for example, the describe of the discuss of the phenotypic and genetic characters, the influence of temperature and the production of DA.
Reviewer 2 Report
Comments and Suggestions for Authors
In this paper, the authors investigated the physiological, morphological, and genetic variability of 40 strains of Pseudo-nitzschia seriata. The authors exposed the strains to three different cold temperatures (2°C, 6°C, 10°C) and did phenotypic assays to measure growth rates, toxin content, and production rates. The results showed extensive intraspecific variation in physiological traits—especially in toxin content and growth rate—while morphological and genetic variation remain relatively low.
It is always a delight to see work proposing the importance of intraspecific genetic variation and how this can play a pivotal role in population dynamics, such as resilience to rising ocean temperatures. The potential for increased toxicity of future blooms is certainly noted in other literature for other cyanobacterial species. and potentially lead to increased toxicity in future blooms.
The manuscript would benefit from more concise framing (at times it feels a bit long). There should also be greater discussion of ecological and public health implications, and deeper contextualization within the HABs literature. For instance, what are the downstream risks to ecosystems, fisheries, and human health? A broader interpretation of what this variation means for bloom forecasting, risk management, and food web dynamics would enhance the paper’s impact. While intraspecific variation is a central theme, the relevance to other Pseudo-nitzschia species or other harmful algae is left implicit and could be more explicitly addressed.
Some references in other study systems that could be leveraged to elevate the implications of your findings for risks to ecosystems, fisheries, and human health, as well as conducting risk assessments are provided below. Microcystis, for instance, is a well-studied organism and offers lots of lessons for water management that could be leveraged in your paper as a case in point:
-Dick, G. J., Duhaime, M. B., Evans, J. T., Errera, R. M., Godwin, C. M., Kharbush, J. J., ... & Denef, V. J. (2021). The genetic and ecophysiological diversity of Microcystis. Environmental Microbiology, 23(12), 7278-7313. https://doi.org/10.1111/1462-2920.15615
-Shahmohamadloo, R. S., Rudman, S. M., Clare, C. I., Westrick, J. A., Wang, X., De Meester, L., & Fryxell, J. M. (2024). Intraspecific diversity is critical to population-level risk assessments. Scientific Reports, 14(1), 25883.
-Shahmohamadloo, R. S., et al. (2023). Diseases and disorders in fish due to harmful algal blooms. In Climate Change on Diseases and Disorders of Finfish in Cage Culture (pp. 387–429). https://doi.org/10.1079/9781800621640.0010
-Xiao, M., Li, M., & Reynolds, C. S. (2018). Colony formation in the cyanobacterium Microcystis. Biological Reviews, 93(3), 1399-1420. https://doi.org/10.1111/brv.12401
Reviewer 3 Report
Comments and Suggestions for Authors
In this study, authors studied about the thermal variation and toxin content of 40 cold water specie of diatom Pseudo-nitzschian seriata. Manuscript is well written, grammatically satisfactory, scientifically sound, results are well interpreted and cited. Figures are presented in good resolution. But some minor suggestions are required to address:
- From which locality, this specie was collected for analyses?
- Samples were collected in 2020 and now it is 2025. Why samples were collected so earlier?
- References are relevant but most of the references are old. Please cite some relevant latest references of the respective journal too.
Reviewer 4 Report
Comments and Suggestions for Authors
The authors have studies 40 species of P. seriata and explored their growth pattern, toxin production, morphological variation and genetic variation at three different temperatures to mimic the situation of global warming affecting the rise of ocean temperatures in the arctic regions. Rise in temperature may lead to higher domoic acid production by the algal bloom leading to higher toxicity of the ocean. A general trend of increasing toxin content with increasing temperature was seen in both growth phases, and most often with a significant correlation. The highest DA content was measured at 10 ï‚°C (9.84 pg DA cell-1). In the exponential growth phase, the mean cellular DA production rates (pg DA cell-1 day-1) in exponential phase were significantly higher at 10 degree C than at the two lower temperatures (p < 0.05; Figure 4), whereas in stationary phase, mean DA production rates did not differ among temperatures (p > 0.1). Variation in the four morphometric characters was compared with previous descriptions of P. seriata, P. australis and P. obtuse. When comparing especially P. seriata and P. australis data, it was evident that ranges of most of the characters (cell width, fibulae density and striae density) were overlapping, confirming their close evolutionary relationship, whereas poroid density was consistently lower in P. australis. Testing the physiological response of 40 strains of P. seriata to three temperatures (2 °C, 6 °C and 10 °C) revealed substantial variation in GR and DA content, which emphasize the importance of considering intraspecific when interpreting physiological responses to environmental factors. Results indicate higher growth potential and higher toxin content at 10 °C, potentially posing a risk of future HABs occurring in warming waters. On the contrary, morphological characters and genetic variation were rather stable parameters. The authors have done an extensive study and the results are very much important in the area of global warming scenario. Higher the temperature, higher is the toxin production which can lead to detrimental effect on the flora and fauna of the marine species. The article is well written and the figures are well presented. The results are also explained very well with a detailed discussion section. I accept the article in its present form with a small minor comment mentioned below that can be done by the authors during galley proof. I congratulate the authors for doing this wonderful study.
In Fig 5 captions, it is mentioned that the poroid pattern in Figure 5 (B) is either @ +2 or 2 + 1 and it is also visible in the Figure. Although the text inside Fig 5 (B) shows only 2 +2 not the 2 +1. Mark the 3 +1 poroid pattern figure also.
Reviewer 5 Report
Comments and Suggestions for Authors
This manuscript discusses Extensive variation in thermal responses and toxin content among 40 strains of the cold-water diatom Pseudo-nitzschia seriata - in a global warming context. An interesting knowledge has been reported. However, the following comments should be addressed before acceptance
Comments
The study uses 40 strains derived from a single water sample. While this controls for environmental variation, it may underrepresent the broader geographic and temporal genetic diversity of P. seriata. Consider including strains from multiple locations or time points to better capture natural variation.
The authors explicitly state that strains were not acclimated to test inherent physiological plasticity. However, without acclimation, results may reflect acute stress responses rather than long-term thermal adaptation. Including a brief acclimation phase or discussing this limitation in more depth would strengthen the interpretation.
Most strains were tested in single replicates except three strains used in triplicate. This limited replication may impact the robustness of inter-strain comparisons. Including technical or biological replicates could provide more reliable variance estimates
The claim that poroid row number may be influenced by temperature is interesting. However, this hypothesis is mentioned briefly without experimental support. Including temperature-dependent morphotype ratios or referencing previous studies would strengthen this point.
Round 2
Reviewer 2 Report
Comments and Suggestions for Authors
The authors have thoughtfully and carefully revised the manuscript. It is sufficient and acceptable for publication.